# On-site growth of perovskite nanocrystal arrays for integrated nanodevices

Patricia Jastrzebska-Perfect [1,2], Weikun Zhu [2,3], Mayuran Saravanapavanantham[1,2], Zheng Li [1,2], Sarah O. Spector [1,2], Roberto Brenes [1,2], Peter F. Satterthwaite [1,2], Rajeev J. Ram [1,2] & Farnaz Niroui [1,2] ✉

Despite remarkable progress in the development of halide perovskite materials and devices, their integration into nanoscale optoelectronics has been hindered by a lack of control over nanoscale patterning. Owing to their tendency to degrade rapidly, perovskites suffer from chemical incompatibility with conventional lithographic processes. Here, we present an alternative, bottom-up approach for precise and scalable formation of perovskite nanocrystal arrays with deterministic control over size, number, and position. In our approach, localized growth and positioning is guided using topographical templates of controlled surface wettability through which nanoscale forces are engineered to achieve sub-lithographic resolutions. With this technique, we demonstrate deterministic arrays of CsPbBr$_3$ nanocrystals with tunable dimensions down to <50 nm and positional accuracy <50 nm. Versatile, scalable, and compatible with device integration processes, we then use our technique to demonstrate arrays of nanoscale light-emitting diodes, highlighting the new opportunities that this platform offers for perovskites' integration into on-chip nanodevices.

Due to their superior optoelectronic properties, lead halide perovskites (APbX$_3$ where A = CH$_3$NH$_3^+$, CH(NH$_2$)$_2^+$, Cs$^+$ and X = Cl$^-$, Br$^-$, I$^-$), have demonstrated their promise in applications such as solar cells[1,2], light-emitting diodes[3], lasers[4], photodetectors[5,6], memristors[7,8], and single photon sources[9,10]. The ability to precisely pattern these materials at the nanoscale with high spatial resolution and positioning accuracy can extend these prospects to on-chip integrated nanodevices. However, prone to degradation, these materials are not compatible with typical top-down fabrication processes[11]. To alleviate this limitation, bottom-up strategies in which crystallization is localized with control over size, shape, and spatial order are promising[11,12]. Confined growth has been demonstrated using patterned templates, where physical topographical scaffolds and/or chemically defined selective surface wetting guides precursor solution localization to initiate site-specific nucleation and growth[13–16]. In other techniques,

patterned deposition of precursor solution is achieved using inkjet printing[17] or polymer pen lithography[18]. While the latter, which employs a scanning probe method, has demonstrated controlled growth at the nanoscale[18], other past developments have more commonly been limited to micron-sized patterning. However, miniaturization to the nanoscale is essential for on-chip nanodevices. To successfully integrate a high-density of optimal devices (in terms of, for example, device coupling efficiencies[19]), this size control must also be complemented by nanoscale precise positioning for accurate alignment of the nanocrystal to other device features[16,20]. In the current bottom-up strategies, however, the desired sub-lithographic precision cannot be achieved as the positioning accuracy is dictated by the area of the confined precursor solution, set by the resolution limit of the patterning technique[14,18]. In existing designs, as the solvent dries, the nanocrystal is positioned stochastically within the original confining

[1]Department of Electrical Engineering and Computer Science, Massachusetts Institute of Technology, Cambridge, MA 02139, USA. [2]Research Laboratory of Electronics, Massachusetts Institute of Technology, Cambridge, MA 02139, USA. [3]Department of Chemical Engineering, Massachusetts Institute of Technology, Cambridge, MA 02139, USA. ✉e-mail: fniroui@mit.edu

area. Thus, alternative strategies are needed. The resulting techniques must simultaneously be scalable and high-throughput while also compatible with the device fabrication steps, such that integration of the perovskite nanocrystals into functional electronic and optical systems can be facilitated.

Here, we introduce a platform for bottom-up, patterned growth of arrays of halide perovskite nanocrystals with control over size, number, and position, while ensuring compatibility with device integration processes. Our approach uses a lithographically patterned topographical template composed of wells with asymmetric surface wettability to confine the nanocrystal growth. We use template geometry and surface wetting to induce local directional forces that further guide nanocrystal positioning during the growth process to resolutions beyond the conventional, lithographically established limits. With this technique, we demonstrate scalable arrays of cesium lead bromide (CsPbBr$_3$) nanocrystals with tunable dimensions down to <50 nm and <50 nm positional accuracy, surpassing those previously reported. Through a systematic study, we investigate the influence of template shape and wetting properties on the number, size, and placement of the formed nanocrystals. We then use our technique to demonstrate arrays of electrically driven nanoscale light-emitting diodes (LEDs), highlighting the new opportunities that may be offered through this platform for perovskites' integration into on-chip nano-optoelectronics.

## Results and discussion
### Scalable and deterministic on-site growth of halide perovskite nanocrystals

Our process for on-site growth of lead halide perovskite nanocrystals with deterministic placement is depicted in Fig. 1a. Core to our approach is a topographical template with wells of controlled shape and surface wettability that selectively confine the perovskite precursor solution and locally guide the growth and positioning process. The template consists of partially wetting wells bored into a non-

wetting surface, fabricated through direct electron-beam lithography in hydrogen silsesquioxane (HSQ) resist or lift-off patterning of evaporated SiO$_2$, as schematically outlined in Supplementary Fig. 1. To achieve the desired asymmetric surface wetting, we implement a two-step surface functionalization with self-assembled fluorinated molecular layers. In the HSQ scheme, the first functionalization step is performed prior to the development of the electron-beam patterned HSQ template, resulting in a top surface that has a contact angle of ~70° with the perovskite precursor solution. Once the HSQ is developed, the hydrophilic interior of the well is exposed. The wetting properties of the well are then further tuned through a second, shorter, molecular growth step (Fig. 1a(i)). This step helps control the precursor solution distribution within the well and its drying dynamics.

Next, the precursor solution, composed of (1:1) PbBr$_2$:CsBr dissolved in dimethyl sulfoxide (DMSO), is spin-coated onto the template, resulting in local reservoirs of precursor solution inside the wells (Fig. 1a(ii)). The exact shape of the meniscus that forms in each well is determined by the well geometry, and the sidewall contact angle, labeled as $\theta$ in Fig. 1a(iii). As the precursor solution evaporates, the meniscus propagates into the well. When the solubility limit of the precursor solution is locally exceeded, a nanocrystal nucleates and grows until the solution is fully dried. The nanocrystal growth is then completed with an annealing step at 80 °C to remove any residual solvent. An example array of on-site fabricated CsPbBr$_3$ nanocrystals, with lateral dimensions of 38 ± 3 nm, formed inside square-shaped wells with 150 nm sides is demonstrated in Fig. 1b. Using transmission electron microscopy and electron diffraction measurements as shown in Supplementary Fig. 2, we confirm that the resulting nanocrystals are single crystalline. Our approach is also scalable. The fluorescence image in Fig. 1c shows four arrays of 12 × 12 individually placed CsPbBr$_3$ nanocrystals. The resulting nanocrystals display the expected chemical composition which we have confirmed through a combination of energy-dispersive x-ray spectroscopy (EDS) and fluorescence spectroscopy. The representative elemental map in Fig. 1d shows the

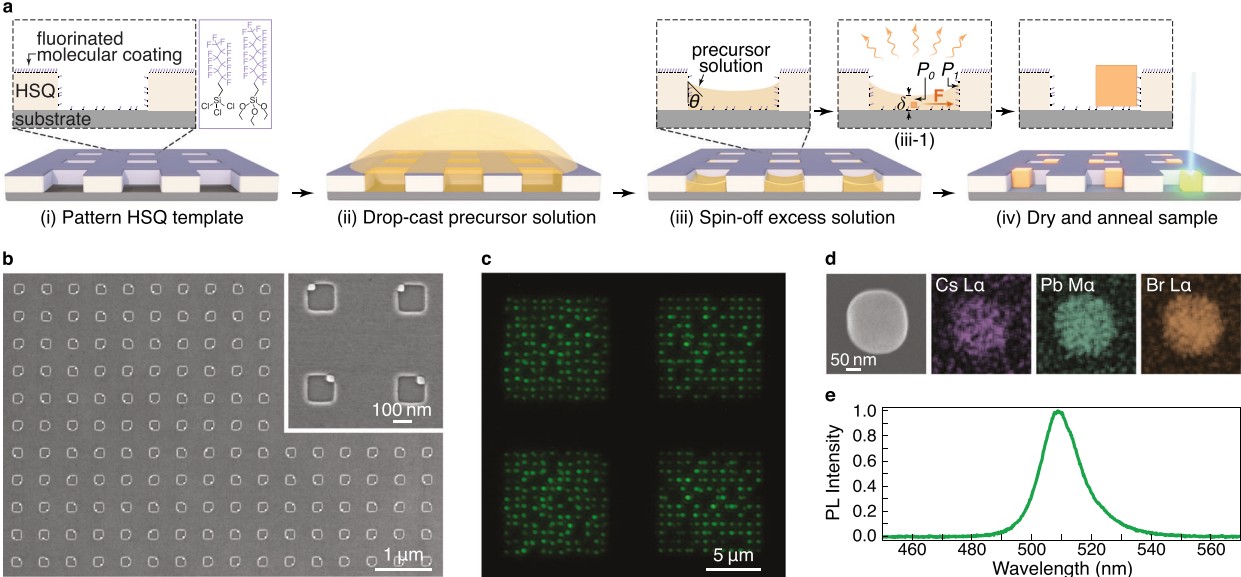

**Fig. 1 | On-site synthesis of halide perovskite nanocrystals with asymmetrically wetting topographical templates. a** Schematic illustration of the nanocrystal synthesis process. (i) Asymmetrically wetting template consisting of the non-wetting top surface and wetting well interior is fabricated using lithographic patterning of HSQ and selective surface treatment with fluorinated molecular assembly. (ii) CsPbBr$_3$ precursor solution is drop-casted on the template. (iii) Excess solution is spun off of the template, leaving precursor solution localized in the wells. (iii-1) In each well, the solution forms a meniscus with a contact angle $\theta$ to the side walls. As the solvent evaporates, a nanocrystal nucleates in the well and continues to grow.

The local variation in meniscus height $\delta$ results in a pressure gradient, leading to a force (F) directed from the highest ($P_0$) to lowest ($P_1$) pressure region which helps guide nanocrystal positioning. (iv) Upon complete solvent removal through the evaporation process and an annealing step, the nanocrystal formation is complete. **b** SEM image of 150$^2$ nm$^2$ square wells containing CsPbBr$_3$ nanocrystals patterned on Si substrate. Inset shows wells at higher magnification. **c** Fluorescent micrograph of four 12 × 12 arrays of 250$^2$ nm$^2$ wells patterned on SiO$_2$/Si substrate. **d** SEM image and corresponding EDS maps of a single CsPbBr$_3$ nanocrystal. **e** Photoluminescent (PL) spectrum of nanocrystals formed in 250$^2$ nm$^2$ well.

uniform distribution of Cs, Pb, and Br, and the 508 nm photo-luminescence emission peak measured in Fig. 1e is consistent with the bandgap range expected for this halide perovskite composition[21,22]. The process is also versatile and can be extended to other types of perovskites with an example of $CH_3NH_3PbBr_3$ demonstrated in Supplementary Fig. 3.

As observed in Fig. 1b, our approach enables the deterministic placement of the nanocrystals confined to the wells. Within the wells, however, the nanocrystals are stochastically located at the four corners. This is evident in Fig. 2a where the position of nanocrystals cumulated across 96 square wells show $25 \pm 7\%$ of the nanocrystals positioned at each corner. During evaporation, a meniscus forms within the 22 nm tall well. Due to interactions with the underlying solid surface, the pressure experienced at the meniscus minimum differs from that experienced in the bulk[23]. This pressure difference, referred to as the disjoining pressure ($P_d$), scales inversely with the liquid thickness ($\delta$), given by $P_d = A_H/(6\pi\delta^3)$, where $A_H$ is the Hamaker constant between the liquid and the underlying surface[23,24]. As a result, with the liquid thickness varying across the meniscus, a pressure gradient emerges across its profile in the well. Assuming no convection or diffusion, the pressure gradient will induce a directional force to guide the nanocrystal from the region of lowest meniscus thickness to the highest, hence the corners, in this case[25]. Given that a square well yields a symmetric meniscus, the nanocrystals are equally likely to reside at any of the four corners, as we have experimentally observed. In this case, the placement precision is limited by the size of the well. The precision can only be improved by decreasing the well size. However, the size is limited by the resolution of the lithographic technique. Further, depending on the surface wettability, small wells can remain largely unfilled and hence without a nanocrystal, as shown in Supplementary Fig. 4. Thus, an alternative approach, beyond tuning the well dimensions, is needed to enhance the placement precision. Such sub-lithographic precision is important for the aligned integration of nanocrystals with the nanostructured elements needed for device development.

## Sub-50 nm placement control with an asymmetric meniscus

In the square well design discussed above, the meniscus adopts a symmetric profile, so a nanocrystal is equally likely to reside in any of the four corners. However, with an asymmetric meniscus, a directional pressure gradient can be formed, allowing for precise placement of the nanocrystal at a particular location. The meniscus shape can be tuned using the contact angle between the precursor solution and well sidewalls, and the well geometry. When the contact angle is smaller than a critical angle $\theta_c$, the meniscus changes from spanning the entire well, pinned at the sidewalls, to having some liquid remaining at the corners as it propagates into the well[26]. An example is shown in Fig. 2b where the contact angle of DMSO to the sidewalls of a square well, whose critical angle is 45°, changes from 50° to 25°. The menisci at the corners are referred to as arc menisci, and the central region is the main terminal meniscus[27]. Depending on the corner geometry, the critical angle for the arc meniscus formation varies. A detailed discussion is included in the Supplementary Note 1. Thus, by using an asymmetric well, an asymmetric meniscus can be obtained with different amounts of liquid trapped at each corner. This is illustrated in the triangular well of Fig. 2b.

To investigate the effect of meniscus asymmetry on the nanocrystal positioning, we fabricated wells of different geometries as shown in Fig. 2a. These include a square, an equilateral triangle, and two isosceles triangles with corner angles (44°, 68°, 68°) and (20°, 80°, 80°). The critical contact angles associated with these well designs are summarized in Supplementary Table 1. To allow direct comparison between the different geometries, the wells are made to support an equal volume of the precursor solution. For this, the well dimensions are selected to yield a constant area corresponding to that of a

150 nm × 150 nm square, while maintaining a constant height of 22 nm for all samples. The wells are further made to have a contact angle of 50° to the precursor solution, through a second functionalization step. This contact angle is controlled by varying the duration of the fluorination step, an example of which is shown in Supplementary Fig. 5a. Nanocrystals, $38 \pm 3$ nm in lateral dimension, are obtained for all the well geometries, which is as expected given the wells' constant volume (Fig. 2c). Depending on the geometry, instances of multiple nanocrystals per well are also observed. Notably, an asymmetric well geometry favors the formation of single crystals with the yield improving to 80% for the (20°, 80°, 80°) template (Fig. 2d).

The positions of nanocrystals formed in over 96 wells of each shape are plotted in Fig. 2a. As with the case of the square well, the symmetry in the equilateral triangle well results in near equal ($33 \pm 9\%$) distribution of the nanocrystals at the three corners. The nanocrystal distribution in the (44°; 68°, 68°) and (20°; 80°, 80°) triangles are (64; $18 \pm 6\%$) and (86; $7 \pm 0\%$), respectively, where distributions for equivalent angles are represented as their mean ± standard deviation. In the triangular geometry, it is clear that the asymmetry favors the nanocrystal positioning at the smallest angle. If considering only the population of wells with single nanocrystals, then 67% and 89% of the nanocrystals are deterministically placed in the 44° and 20° corners, respectively. As the smallest corner angle is reduced, an improved positional accuracy of <50 nm is achieved (Fig. 2d). Supplementary Note 2 details the evaluation of the positional accuracy, which measures the particle distance from the corner in which it is expected to be located as per the balance of forces. If considering only wells with single nanocrystals, an improved positional accuracy <10 nm is measured for the (20°, 80°, 80°) well. This demonstrates the ability of meniscus engineering to achieve nanocrystal positioning with sub-lithographic resolution, beyond that offered by the well dimensions alone.

The preferential positioning stems from the pressure gradients leading to a force dominating in a desired direction. This directionality comes from the asymmetry engineered into the meniscus with the largest effective force experienced towards the corner with the largest volume of liquid in the case of the triangular well—that is, the arc meniscus with the largest area. This effect is explained in detail in Supplementary Note 3 and can be visualized using the schematic illustrations of the four experimentally implemented template geometries in Fig. 2e. Here, we also show the calculated arc meniscus areas for each corner angle as a function of the sidewall contact angle. Details of calculations, which were based on expressions for arc meniscus radii[28], are provided in Supplementary Note 4. The largest arc meniscus area is observed for the 20° corner. In this geometry, the experimentally implemented contact angle of 50° is selected to satisfy $\theta_{c1} \leq \theta < \theta_{c2}$ where $\theta_{c1} = 50°$ corresponds to the critical angle for the 80° corners and $\theta_{c2} = 80°$ for the 20° corner. As a result, an arc meniscus only forms at a smaller angle. This maximizes the extent to which the nanocrystal driving force is preferentially directed to one corner, compared to the other geometries, allowing for higher yield deterministic nanocrystal positioning, as is experimentally observed.

## Controlling nanocrystal size and number

In addition to extreme miniaturization to <50 nm crystal dimensions and precise positioning with <50 nm accuracy, our platform can also control the nanocrystal size. One approach is through changing the precursor solution concentration where a higher concentration yields larger nanocrystals as shown in Supplementary Fig. 6. The extent of tunability through this approach is, however, limited as the concentration needs to be maintained close to the saturation regime to facilitate the crystal formation process. The nanocrystal size can alternatively be controlled by changing the volume of the trapped precursor solution through modifying the well size. This can be readily implemented through the lithographic patterning of the template and

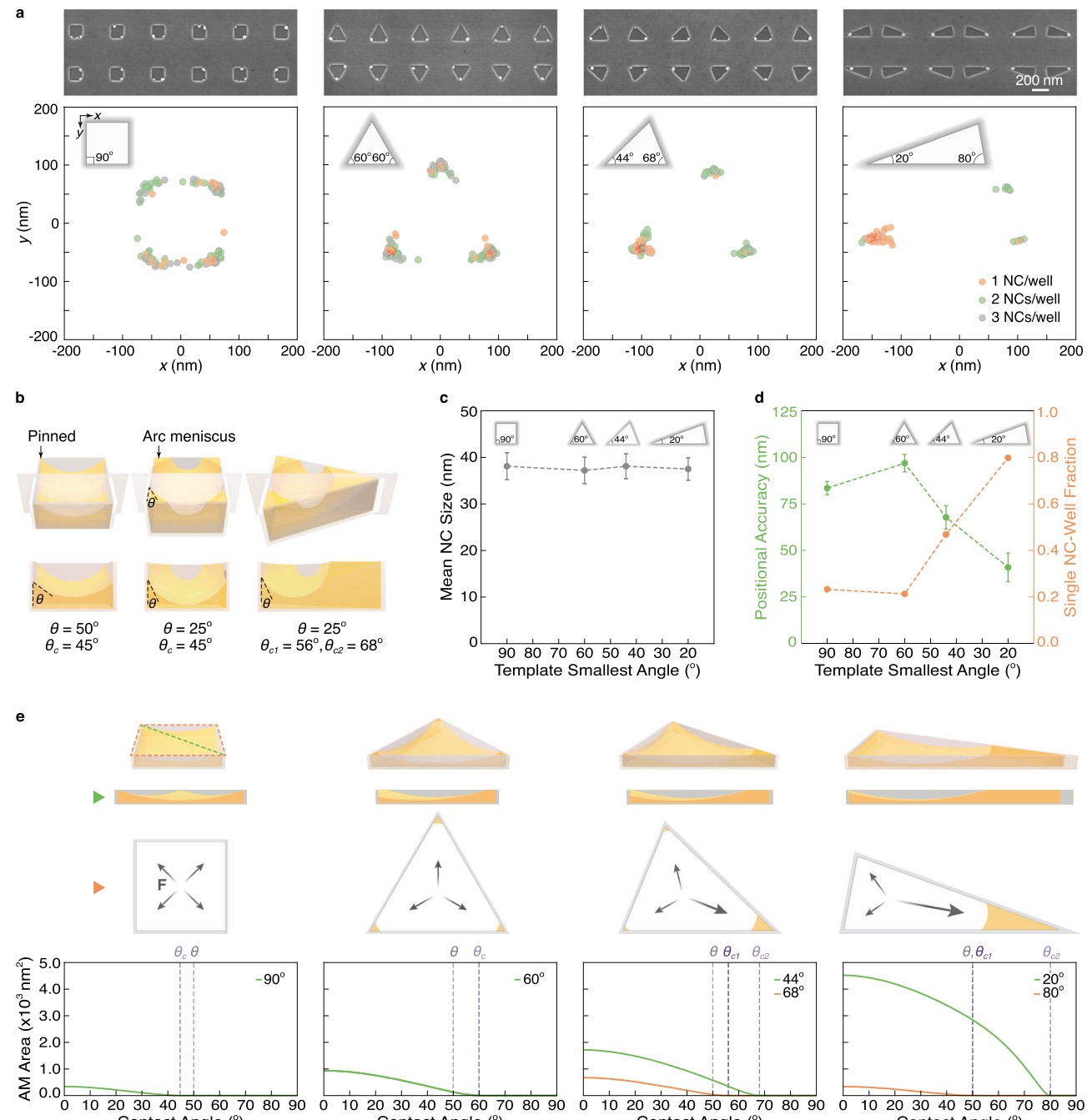

**Fig. 2 | Deterministic nanocrystal placement with an asymmetric meniscus.**
**a** Representative SEM images (top), and plots of nanocrystal (NC) position with respect to well centroids cumulated over 96 wells (bottom), for templates of four different well shapes, and common volume ($150^2$ nm$^2$ area and 22 nm thickness) and sidewall contact angle ($\theta = 50°$). From left to right, well shapes are square, equilateral triangle, (44°, 68°, 68°) triangle, and (20°, 80°, 80°) triangle. The colors indicate the presence of different numbers of nanocrystals within the well. **b** Three-dimensional (top) and cross-sectional schematic views (bottom) of menisci in polygonal wells. From left to right, wells are (i) square with $\theta > \theta_c$, (ii) square with $\theta < \theta_c$, and (iii) triangle with $\theta < \theta_{c1}, \theta_{c2}$, where $\theta$ is solvent-sidewall contact angle, and $\theta_c$ is critical contact angle. **c** Nanocrystal size for the four different well shapes considering single nucleations only (mean ± standard deviation). **d** Positional accuracy (mean ± standard error) and a fraction of wells containing one nanocrystal per well for the four different well shapes. **e** Meniscus profiles in experimentally studied wells, where meniscus minimum is set at 20% of well height, and arc meniscus area is analytically determined. Menisci are drawn to scale with experimental dimensions (top). Arc meniscus (AM) area as a function of solvent–sidewall contact angle for each corner angle, where critical angles are indicated with dashed lines (bottom).

importantly allows for nanocrystals of varying dimensions to be fabricated in parallel on the same substrate, with a larger well expected to yield a larger nanocrystal.

To confirm this trend, we fabricated (20°, 80°, 80°) triangular wells with nine different volumes as shown in Fig. 3a. These are composed of wells with three different template thicknesses (22, 28, and 36 nm), and three different areas ($150^2$, $200^2$ and $250^2$ nm$^2$). We used a

combination of SEM imaging to probe the lateral dimensions of the nanocrystals, and an atomic force microscope (AFM) to identify their heights. It should be noted that the height of the nanocrystals is set by the well thickness and varies from their lateral dimensions. The results measured over 30 wells of each volume, containing single nanocrystals, are plotted in Fig. 3b, demonstrating an increase in the nanocrystal size with the volume of the well in which it was formed. Here,

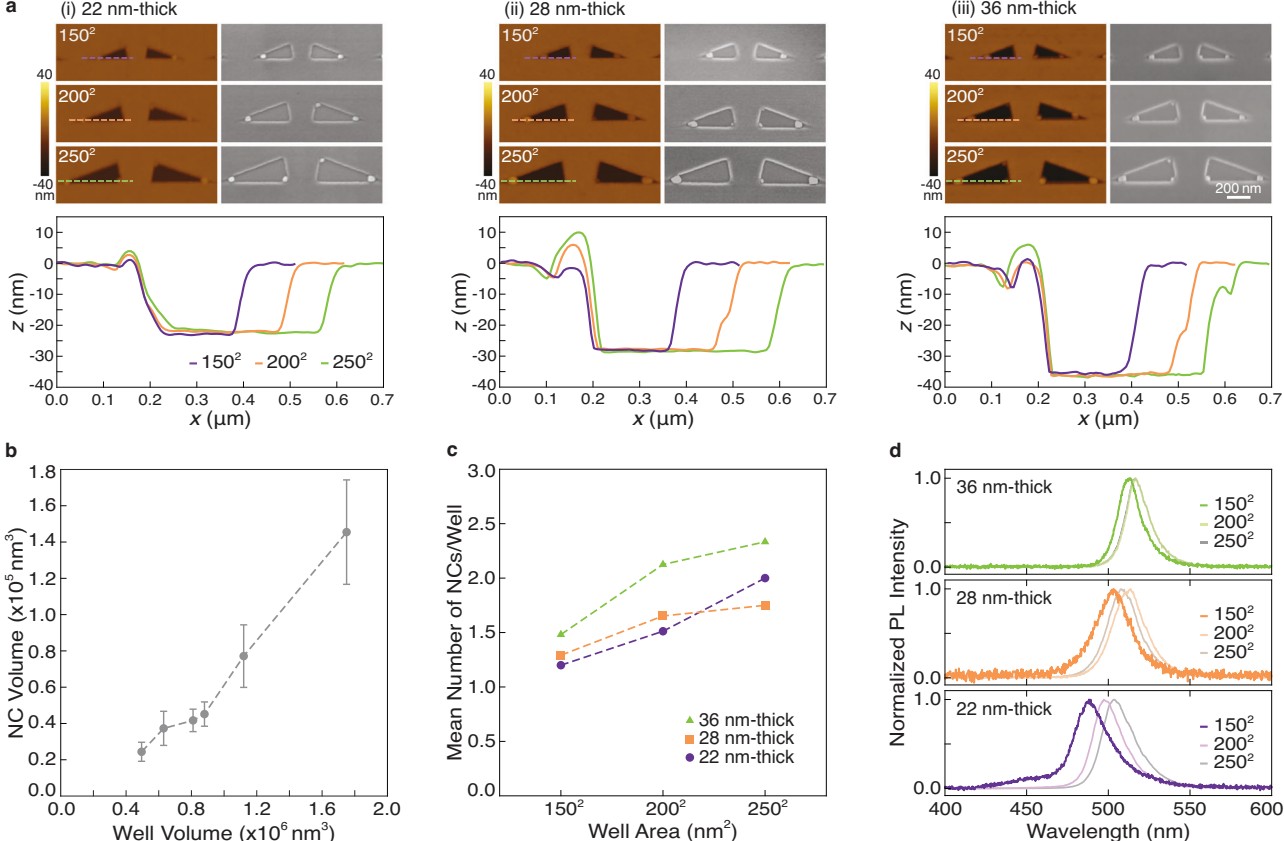

**Fig. 3 | Well dimensions control size and number of nanocrystals. a** AFM images (left), correlated SEM images, and AFM line traces (bottom) for representative (20°, 80°, 80°) triangle wells of $150^2$, $200^2$, or $250^2$ nm² areas, for (i) 22, (ii) 28, or (iii) 36 nm well thicknesses. **b** Nanocrystal (NC) volume (mean ± standard deviation) as a function of well volume for wells containing single particles. **c** Number of particles per well as a function of well area for three template thicknesses. **d** The mean photoluminescence (PL) spectra for nanocrystals formed in wells of varying areas and thicknesses with the darker curves presenting wells with primarily single nucleation.

the nanocrystal size is represented as its volume, and the lateral dimensions and height associated with each point are summarized in Supplementary Table 2 for clarity.

For each template thickness, as the well size increases, we also observe an increase in the fraction of wells with multiple nanocrystals (Fig. 3c). In such a well, the cumulative size of all the nanocrystals matches that of an individual crystal formed in a same-sized well (Supplementary Fig. 7). This confirms that the wells confine the same volume of precursor solution as expected, and the number of the nucleation events is driven by the varying drying dynamics where a shallow meniscus with a larger surface area can accommodate multiple nucleation sites. It follows then, that changing the well volume by scaling the thickness of the template, rather than the lateral dimensions alone, will push the threshold at which instances of multiple nanocrystal nucleations become prevalent. This is illustrated in Fig. 3c where the $150^2$ nm² well area that supports one nanocrystal per well at the lowest studied thickness, still predominantly supports one at greater thicknesses. Without a change in the well size, the number of nanocrystals can also be controlled by adjusting the precursor solution contact angle to the well. As shown in Supplementary Fig. 5b, the frequency of multi-nucleation events reduces with a lower contact angle. Thus, through a balance between the well size and surface wettability, the size and number of nanocrystals can be controlled.

By controlling size, the nanocrystals' optical properties can be modified. The photoluminescence (PL) emission spectra were collected for the nanocrystals formed in the different well volumes. The spectra are shown in Supplementary Fig. 8 and their cumulated averages for each condition are plotted in Fig. 3d. To elucidate the size-dependent PL trend, the wells that yield the highest fraction of single

nanocrystals ($150^2$ nm² wells for each thickness) are focused on. Consistent with prior reports[18,21,22], Fig. 3d shows that the peak photoluminescence emission blue-shifts to higher energies as the nanocrystal size decreases, despite crystal dimensions exceeding dimensions needed for quantum confinement. This blue-shift effect has been attributed to photon reabsorption modulating the relative intensities of the multiple emission modes that exist due to defects at the band edges[18].

In allowing the scalable, controlled formation of perovskite nanocrystals, our platform provides a high-throughput approach to examining how the nanocrystals' physical and chemical features affect their optoelectronic properties. For example, the effects of nanocrystal size, orientation, and morphology on PL emission may be elucidated, which are speculated to contribute to the current nonuniformities observed in the emission characteristics. Furthermore, nanocrystal composition, ligands, surface passivation, and growth conditions may be varied to optimize the performance of the nanocrystals, whose photoluminescence quantum yield is estimated in Supplementary Note 5.

## Demonstrating arrays of on-chip nanoscale light-emitting diodes

To highlight the new device integration opportunities offered by our platform, we next demonstrated arrays of $CsPbBr_3$ nanoscale light-emitting diodes (nanoLEDs). Such nanoLEDs have applications in enhancing resolution in the near-eye displays used in augmented and virtual reality[29,30], lensless microscopy[31–33], on-chip optical communication[34,35] and computing[36], and realization of quantum light sources[37,38]. Even though larger area halide perovskite LEDs have been

investigated to leverage the color purity, spectral tunability, and solution processibility of perovskites[3,39], nanoscopic miniaturization of perovskite LEDs has yet to be realized due to the challenge in patterning the emissive layer with high-resolution and deterministic spatial order. Our approach addresses this challenge and enables scalable self-aligned integration of perovskites into dense arrays of functional nanoLEDs.

Our device structure is schematically shown in Fig. 4a and a representative cross-sectional SEM image is shown in Fig. 4b. First, we use our technique to pattern arrays of $CsPbBr_3$ nanocrystals with the desired size and spatial arrangement on an indium tin oxide (ITO) electrode. Here, a modified fabrication technique is used to form the $SiO_2$ topographical template as per the process schematically shown in Supplementary Fig. 1b and discussed in detail in the "Methods" section. In this example, the templates have a teardrop shape yielding a meniscus whose profile is governed on one end by the semicircle, and the other by the corner angle of the wedge[27] (Fig. 4c). In this design, as the arc length of the semicircle is greater than that of the arc meniscus, the asymmetric meniscus design induces an effective force directed towards the rounded corner where the nanocrystal is preferentially placed. This is evident in the SEM micrographs in Fig. 4d. Once the nanocrystals are formed, a <5 nm layer of poly(methyl methacrylate) (PMMA) is deposited over the substrate. Finally, the 2,2′,2″-(1,3,5-benzinetriyl)-tris(1-phenyl-1-H-benzimidazole) (TPBi) electron-transport layer and the LiF/Al top contact are thermally evaporated[40,41]. In this design, the wells define the individual pixels, while the $SiO_2$ and PMMA help limit shorts and leakage current between the top and bottom electrodes and eliminate background emission from the TPBi[37,41].

Using the designs in Fig. 4d, we fabricated arrays of 600–2400 nanoLEDs connected in parallel. An example array of 400 devices is shown in the electroluminescence (EL) image in Fig. 4e. In this example, a device is made using a teardrop well (namely a semicircle merged with a (20°, 80°, 80°) triangle of 1000 nm² area), which is formed within a 57 nm-thick $SiO_2$ layer. The device design yields perovskite nanocrystals of $214 \pm 23$ nm in the lateral dimensions. An example current–voltage characteristic of the devices is included in the Supplementary Fig. 9a. The average EL emission measured is centered at 521 nm, representative of $CsPbBr_3$, with full-width at half-maximum of

20 nm (Fig. 4f). The background emission from TPBi evident under optical excitation of the LED with a 405 nm laser is not observed in the EL spectrum emission (Supplementary Fig. 9b), highlighting the suitability of our design for preventing leakage and background emission. We estimated the average EL power of a single LED to be -0.6 fW at 10.5 V. The details of this estimate are provided in Supplementary Note 6, along with an analysis of the external quantum efficiency. In the demonstration shown here, the particular size of the nanoLED is selected to enable easier optical characterization. However, as we demonstrated previously, the size and spatial arrangement of the individual wells and hence the nanoLEDs can be readily changed through the template design. Additionally, by controlling the nanocrystal composition and surface passivation, or utilizing other device architectures, LEDs with tunable emission wavelength and improved efficiencies may be achieved.

In conclusion, we have demonstrated a high-throughput and scalable approach for on-site synthesis of perovskite nanocrystals with controlled size and deterministic spatial arrangements. Using a topographical template with asymmetric surface wetting, we confine the precursor solution to localize the crystal growth and engineer the nanoscale forces to position the nanocrystal with <50 nm accuracy. By controlling the precursor solution concentration and the template dimensions, we further show that the size of the nanocrystals can be controlled down to <50 nm. In addition to enabling high-throughput studies of perovskites at the nanoscale to elucidate the structure–chemistry–function relationships, this approach and its compatibility with other fabrication techniques further provide a platform that helps extend perovskites' applications to on-chip nanoscale devices. We highlight this opportunity by demonstrating $CsPbBr_3$ nanoscale LEDs formed with deterministic placement and spatial arrangement. However, the prospects may be extended to a diverse set of device platforms utilizing perovskites' superior optoelectronic properties such as on-chip lasers, memristors, and quantum light sources.

## Methods
### Template fabrication
The two template fabrication schemes depicted in Supplementary Fig. 1 are described here. In both schemes, substrates were cleaned

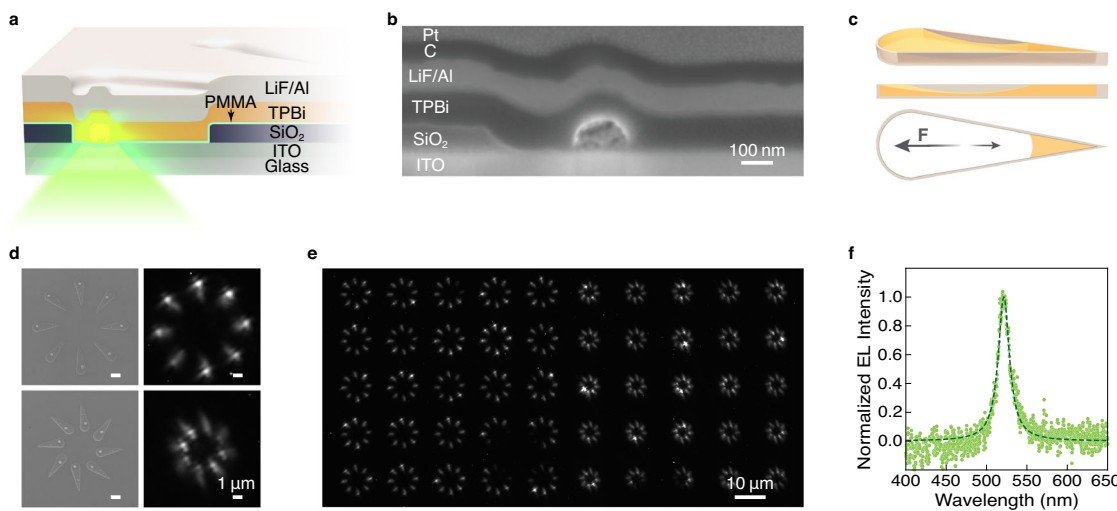

**Fig. 4 | Deterministic growth of perovskite nanocrystals enables on-chip nanoscale LEDs. a** Schematic illustration of the $CsPbBr_3$ nanoLED formed using teardrop wells in a $SiO_2$ topographical template. Note that the encapsulation is not visualized here. **b** Representative cross-sectional SEM image of a nanoLED. The C and Pt layers are deposited to protect the device during focused ion beam (FIB) cross-sectioning. **c** Meniscus profile in a teardrop well used as a nanoLED pixel. The meniscus results in an effective force positioning the nanocrystal in the semicircular end. **d** Representative SEM images of nanocrystals formed in teardrop wells on a silicon substrate, where each teardrop well consists of a semicircle merged with a 1000² nm² (20°, 80°, 80°) triangle (left). Electroluminescent (EL) images of nanoLEDs formed using the same design of wells taken under 10.5 V biasing voltage (right). **e** EL image of 400 nanoLEDs, arrayed in the two orientations shown in (**d**). **f** EL spectrum of the nanoLED array in (**e**).

sequentially with acetone under sonication and boiling isopropanol prior to use. Silicon (Si) substrates were used for samples for SEM analysis, and 1 μm silicon dioxide-on-silicon (SiO$_2$/Si) substrates were used for those for optical characterization. Patterned 145 nm-thick ITO on glass substrates (ITO/glass), purchased from Thin Film Devices Inc., were used for nanoLED fabrication.

**HSQ template fabrication.** Substrates were treated under oxygen plasma for 3 min to acquire a hydrophilic surface. Hydrogen silsesquioxane (HSQ, Dow Corning) was spin-coated onto the substrates to the desired thickness (22 nm unless otherwise specified). HSQ was then exposed using an Elionix FS-125 electron-beam lithography system operating at a 2 nA beam current and 125 keV accelerating voltage, with a dose of 11,000 μC/cm$^2$. After e-beam exposure, the surface was functionalized with trichloro(1H,1H,2H,2H-perfluorooctyl)silane (PFTS, MilliporeSigma) or 1H,1H,2H,2H-perfluorodecyltriethoxysilane (PFDTES, MilliporeSigma) under vacuum in a desiccator containing 30 μL of PFTS for 10 min, or 30 μL of PFDTES and 90 μL of deionized (DI) water for 9-12 h. The exposed sample was developed in a salty developer solution of 1 wt% NaOH and 4 wt% NaCl for 1 min, then rinsed with gentle stirring in DI water three times for 2 min each, and dried under a stream of nitrogen. To ensure the molecular layer functionalization and the resulting surface wetting remains consistent across samples, the volumes of developing and rinsing solutions were standardized, and fresh developing and rinsing solutions were used for each sample. After developing, the same molecule (PFTS or PTDTES) was briefly deposited by leaving the samples under a vacuum in a desiccator for 5–10 s for PFTS, or 10-40 min for PFDTES, and the contact angle made by DMSO on the substrate was measured.

**SiO$_2$ template fabrication.** Clean substrates were used without additional surface treatment. A bilayer resist stack was prepared by spinning poly(methyl methacrylate) e-beam resist (PMMA resist, 950K A2, Kayaku) at 1000 rpm for 60 s, baking the substrates on a hot plate for 2 min at 180 °C, then spinning 2% HSQ at 1500 rpm for 60 s. HSQ was exposed and developed using the same conditions as above. Next, PMMA was etched with oxygen plasma for ~13 min using the HSQ mask, resulting in PMMA/HSQ pillars. Silicon oxide with a thickness of 57 nm was deposited at a rate of 0.5 Å/s on the substrates by e-beam evaporation under $<7 \times 10^{-6}$ Torr base pressure, with oxygen flowing (3 sccm) during the evaporation. Vapor phase growth of PFDTES was performed on SiO$_2$ templates for 9–12 h. Following surface functionalization, liftoff was performed by leaving the samples in acetone for ~2 h, then sonicating for 30 min. Finally, the contact angle of the wells was tuned through a second PFDTES deposition between 0 and 10 min.

**On-site growth of halide perovskite nanocrystals in the template**
A 0.45 M CsPbBr$_3$ precursor solution was prepared by first dissolving lead (II) bromide powder (99.999% PbBr$_2$, MilliporeSigma) in anhydrous dimethyl sulfoxide (DMSO, 99.9%, MilliporeSigma), then adding the PbBr$_2$ solution to a vial containing cesium bromide powder (CsBr, 99.999%, MilliporeSigma). The CsPbBr$_3$ solution was stirred overnight. All solutions were prepared in a nitrogen glovebox. Nanocrystal growth was performed in a nitrogen glovebox immediately after template fabrication. Precursor solution was deposited by spin-coating 15 μL of precursor solution on a ~0.5" by 0.5" chip at 1000 rpm for 30 s, using 1000 rpm/s acceleration. After deposition, samples were allowed to dry under ambient temperature in the glovebox for 30 min, then annealed at 80 °C for 1 min each. After annealing, the samples used in single-particle spectroscopy were immediately encapsulated inside the glovebox by spin-coating 100 μL of polymer-encapsulating solution on the substrate at 5000 rpm for 60 s. The encapsulating solution was a 3% w/v polystyrene (average $M_w$ 35 kDa, MilliporeSigma) in toluene, filtered through a 1 μm PTFE filter.

**NanoLED fabrication**
Templates were formed using the SiO$_2$ template fabrication scheme, and nanocrystals were grown in templates as per the process outlined above. Then, a thin PMMA film was deposited by spin-coating a 0.0015% w/v solution of PMMA (average $M_w$ 100 kDa, MilliporeSigma) in toluene on the templates at 5000 rpm for 60 s, followed by etching in oxygen plasma for 3 s. Next, 60 nm of 2,2',2"-(1,3,5-Benzinetriyl)-tris(1-phenyl-1-H-benzimidazole) (TPBi, Luminescence Technology Co.) was thermally evaporated on the substrates. The top electrodes were patterned by thermally evaporating ~1 nm of lithium fluoride (LiF) followed by ~75 nm of aluminum through a shadow mask[40,41]. LEDs, except those used for cross-sectional imaging, were encapsulated using a piece of glass adhered to the substrate with a UV-curable epoxy (Delo-Katiobond LP655), inside a nitrogen glovebox.

**Contact angle measurements**
Static contact angles were determined by drop-casting 0.5 μL of DMSO on a substrate and imaging the droplet profile. Images were processed using the low bond axisymmetric drop shape analysis (LB-ADSA) plug-in of the open-source software ImageJ[42]. Contact angles were measured in ambient air immediately after molecular growth.

**Optical measurements**
Fluorescence images were acquired using a Nikon Eclipse LV100 microscope with an attached CMOS sensor camera (AmScope, MU1203-FL). The sample was illuminated by a 100 W halogen lamp through a filter cube consisting of a 475/35 nm excitation filter, 530/43 nm emission filter, and 499 nm edge dichroic mirror (Semrock, BrightLines, and FITC-A-Basic-NTE, respectively). Images were obtained using a ×50, NA = 0.80 objective with 2000 ms integration time.

Fluorescence measurements were performed using a custom-built confocal microscope. Samples were excited by a 405 nm pulsed laser (PicoQuant, LDH-P-C-405, <90 ps pulse duration), through a ×100, NA = 0.85 objective, yielding an on-sample spot size ~640 nm in diameter. Laser excitation was directed onto the sample through a 405 nm dichroic filter with 423 nm cut-on wavelength (Chroma, zt405rdc). Sample fluorescence was collected using the same objective and dichroic. Samples were excited at 57 W/cm$^2$ and 10 MHz repetition rate. Nanocrystal spectra were acquired by first confocally mapping the region of interest (i.e. an array of nanocrystals) to identify the locations of nanocrystals, then collecting spectra from individual points on the array. Mapping was performed by scanning the sample using a piezoelectric scanning stage. Emitted light was passed through a 450 nm long-pass filter (ThorLabs, FEL0450) to remove residual excitation light. Photons were detected using a single-photon detecting avalanche photodiode (Micro Photon Devices, PD-050-00E) connected to a single-photon counting module (PicoQuant, PicoHarp 300) and counted using commercial software (PicoQuant, Sympho-Time 64). Fluorescence spectra were collected using a monochromator (Princeton Instruments, SP2300) and a thermoelectrically-cooled CCD (Princeton Instruments, PIXIS 100). Fluorescence spectra were obtained using 5000 ms of integration time.

**Structural characterization**
The procedure used to characterize the structural details of the resulting perovskite nanocrystals is outlined below.

**Nanocrystal positioning.** Templates were imaged using the in-lens detector of a Zeiss Gemini 450 scanning electron microscope (SEM). The position of nanocrystals in wells was determined from SEM images using custom code written in MATLAB. Well centroids were identified by performing morphological operations (dilation, filling, and erosion) on flattened and binarized SEM images, then finding the centroid of each resultant connected region. Nanocrystal centroids were similarly identified. The location of each nanocrystal $(x_i, y_i)$ in a well was then

defined with respect to the centroid of the well, which was set as (0,0). If a well contained multiple nucleations, the data were appropriately tagged for analysis.

**Nanocrystal lateral size.** Templates were imaged at ×60k–100k magnification, depending on well size. Nanocrystals were distinguished from the background by performing morphological operations (dilation, filling, and erosion) on flattened and binarized SEM images and identifying the connected regions. The length (width) of a nanocrystal was defined as the major (minor) axis length of a given connected region. Measurements were converted from pixels to nanometers using the image scale bar.

**Nanocrystal height.** Nanocrystal height was characterized using an atomic force microscope (Asylum Research, Cypher S) operating in AC mode, with a 160AC-NA cantilever. Second-order flattening was performed on regions surrounding each well to remove offset, slope, and bow. Nanocrystal height was defined as the maximum height of a nanocrystal from the bottom of the well.

**Elemental analysis.** Energy-dispersive x-ray analysis (EDS) was performed using an Oxford AZtec 100 EDS detector attached to a Zeiss Gemini 450 SEM. Images were taken at a working distance of 8.5 mm, under 10.00 kV accelerating voltage and 1 nA beam current. Maps were acquired using the AZtecLive software and exported using a binning factor of 2 and smoothing level of 3.

**Cross-sectional imaging.** Focused ion beam (FIB) cross-sectioning of the nanoLED was performed using the FEI Helios 660 (Thermo Fisher Scientific). First, protective layers of carbon (100 nm) and platinum (500 nm) were deposited onto the target device using electron beam-mediated deposition with a gas injection system. Additionally, a secondary layer of platinum (500 nm) was deposited using ion-beam-mediated deposition. Cross-sectioning was then performed using gallium ions. A similar approach was used to prepare a thin lamella for TEM imaging and diffraction measurements. The lamella, which consisted of nanocrystals formed in HSQ wells on a $SiO_2$/Si substrate, was imaged with JEOL ARM 200F scanning transmission electron microscope at 200 kV accelerating voltage.

### NanoLED characterization

LEDs were characterized by probing samples on an inverted confocal microscope. Samples were imaged through the ITO/glass substrate using a ×50, NA = 0.70 objective. Electrical measurements were performed using a Keithley 2602A source measure unit. Electroluminescence images were collected with a thermoelectrically-cooled monochromatic camera (QImaging, EXi-AQA-R-F-M-14-C), under 5 min of integration time. The images were used to estimate the EL power per single LED as per the process outlined in Supplementary Note 6. Electroluminescence spectra were collected using a monochromator (Princeton Instruments, SP2300) and a thermoelectrically cooled CCD (Princeton Instruments, PIXIS 100).

### Data statistics

Data are reported as mean ± standard deviation unless otherwise indicated.

## Data availability

The data that support the findings of this study are available within the paper and its Supplementary Information. All other relevant data are available from the corresponding author upon request.

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

## Acknowledgements

We thank Irmgard Bischofberger and Paul Lilin for insightful discussions. This work was supported in part by the National Science Foundation (NSF) Award DMR-2144136 and the MIT Center for Quantum Engineering (CQE). P.J.-P., P.F.S., M.S., S.O.S. and R.B. acknowledge support from the NSF Graduate Research Fellowship Program under Grant No. 1745302. The focused-ion beam microscopy and transmission-electron microscopy were performed at the Harvard University Center for Nanoscale Systems (CNS), a member of the National Nanotechnology Coordinated Infrastructure Network (NNCI), which is supported by the National Science Foundation under NSF ECCS Award No. 1541959. The fabrication and characterization procedures in this work have been in part carried out using the MIT.nano shared facilities.

## Author contributions

P.J.-P. and F.N. conceived the project, designed the experiments and wrote the manuscript. W.Z. assisted with some rendering of schematics for the figures. P.J.-P. performed the experiments and analyzed the results. W.Z., M.S., P.F.S and F.N. assisted with developing the fabrication of the light-emitting diodes. S.S. helped perform the focused-ion beam and cross-sectional TEM imaging of the nanocrystals and light-emitting diodes. W.Z., M.S., R.B. and F.N. assisted with the optical measurements. Z.L. and R.J.R assisted with measuring and estimating the nanoLED EL power. F.N. supervised the project. All authors discussed the results and contributed to finalizing the manuscript.

## Competing interests

P.J.-P. and F.N. declare that a patent application has been filed covering the process presented in this work. The remaining authors declare no competing interests.
