## [Peer Review File · Nature Communications]

REVIEWER COMMENTS

Reviewer #1 (Remarks to the Author):

The manuscript authorized by Farnaz Niroui et al., shows some interesting results about on-site growth of perovskite nanocrystal arrays and integrated nanodevices. I recommend the authors address the following questions before considering for publication.

1. The authors said that "It is clear that the asymmetry favors the nanocrystal positioning in the smallest angle." However, the nanocrystals do not position in the smallest angle, as shown in Fig. 4d. What is the reason?
2. Is the nanoparticle nanocrystalline or polycrystalline? I suggest the authors provide TEM images.
3. It is suggested that the PL peak shows blueshift with nanocrystal size decrease. However, there is no data about the size of the nanocrystal. There is also no size distribution of the particles. Several words were used to mean "size", such as size (even particle size or nanocrystal size), high, area and volume in the manuscript. It is very confusing. It is better to clearly define about the size of the nanocrystals and the particles (including that in the lateral dimensions and vertical dimension). Is the thickness of particle determined by the well thickness?
4. Is the demonstrated technique general to other perovskite nanocrystals?
5. It looks that some particles are not positioned at the desirable place. Can the authors approach the limit with high placement accuracy?

Reviewer #2 (Remarks to the Author):

The manuscript reports a novel bottom-up template-assisted synthesis of perovskite nanocrystal arrays. In brief, the authors employed topographical templates with controlled surface wettability to guide the growth and positioning of perovskite crystallites. The authors made substantial efforts to understand the underlying principles of growth and attempted to optimize the nucleation sites of the perovskite by modifying the size and shape of the templates. The authors have further demonstrated that the perovskite crystallites can be used to make high yield LED arrays. The level of control is very impressive. The study can potentially make a very nice contribution to the community and to nature communications. Below are some suggestions for further improvements:

1. According to the current results, although the nucleation sites could be controlled within 50 nm, their morphology or orientation does not appear to be uniform, which may have contributed to the non-

uniform PL and EL in the LED arrays. The authors are suggested to conduct a more detailed analysis of the size, morphology variation and their impact on the PL or EL emissions.

2. The EL image, although with a very impressive yield, showed a relatively high degree of nonuniformity. Additionally, there is no discussion of the EL efficiency either. For LED display applications, the uniformity of the pixels is a critical factor that must be considered. It is important for the authors to explore the underlying reasons for such non-uniformity and the possible strategies to address the problem.

3. Lastly, although it is claimed as EL, there is no evidence to support whether the emission is from direct EL from the perovskite crystallites or down-conversion of EL from the TPBi layer. Although this may not affect the apparent device function, it could affect the overall efficiency. The authors are encourage to look into or at least discuss about relevant uncertainties.

Reviewer #3 (Remarks to the Author):

The emerging perovskite materials show great promises for the next generation micro-LED. However, most perovskite LEDs are based on polycrystalline films prepared by spin coating, which need sophisticated lithography steps to get micro-LED. The pattern of nanocrystals is an alternative strategy. But traditional patterning is limited to micron-sized patterning, and the sub-lithographic precision cannot be guaranteed as the patterning area is confined by the precursor solution. Here, in this manuscript the authors reported the integration of perovskite nanocrystal arrays into nanoscale LEDs. Both the size and position of crystals can be well controlled down to <50 nm. I believe the further optimization of this kind of sample preparation methods, their properties, and the integration into nanoscale LEDs will bring in a revolution for LEDs. The developed method in this manuscript will also be beneficial to other on-chip nanodevices. Here are some questions for the improvement of this manuscript:

1. What's the PLQY of the crystals? As well known, the efficiency will decrease as the size of LED decreases. What's the EQE of this device? Could the authors show the data?
2. In Fig. 1c and Fig. 4d, for both PL and EL, the emission intensity is not homogeneously across the whole patterns. For some area, the emission intensity is rather low. What's the reason for that, can it be improved?
3. The architecture of the LED should be optimized. Currently, the crystal is sandwiched between 5nm PMMA and TPBi, there is no balanced injection of electrons and holes. The recombination of electrons and holes in the crystals may also not be efficient.
4. In Fig. 4b, the arc crystals only cover a few area between PMMA and TPBi. However, if the authors use cubic crystals, the coverage may be better, and the EL performance may also be improved.
5. Electrodes of thin-film transistor (TFT) can be precisely controlled into nanoscale. TFT architecture may be more suitable for this small crystal.
6. The authors said they got the first perovskite nanoLED. As I know, there was another report of nanoLED (Nature Photonics, 2022: 16, 284-290).

REVIEWER COMMENTS

Reviewer #1

The manuscript authorized by Farnaz Niroui et al., shows some interesting results about on-site growth of perovskite nanocrystal arrays and integrated nanodevices. I recommend the authors address the following questions before considering for publication.

We thank the reviewer for their comments. We have addressed their questions in detail below.

1. The authors said that “It is clear that the asymmetry favors the nanocrystal positioning in the smallest angle.” However, the nanocrystals do not position in the smallest angle, as shown in Fig. 4d. what is the reason?

The text quoted by the reviewer corresponds to the studies performed with wells having a triangular geometry. Indeed, as indicated by the reviewer, in the case of triangular wells, the asymmetry of the meniscus favors nanocrystal positioning in the smallest angle. The reason for this selective positioning is explained in detail in Supplementary Note 3 and results from the pressure gradient emerging due to the meniscus thickness variation across the well leading to a directional force which in the triangular well is going to be largest towards the small angle corner with the largest area of arc meniscus. However, Fig. 4d uses wells with a teardrop geometry, whose meniscus profile is illustrated in Fig. 4c. The meniscus meets the sidewalls of the well with the contact angle established via surface-functionalization. This means that within a teardrop well, the meniscus demonstrates its maximum thickness both at the arc meniscus and at the hemispherical perimeter. In Supplementary Note 3, we establish that a particle has the minimum energy at the meniscus maximum. Since the arc length of the hemispherical perimeter is greater than the arc length of the arc meniscus, the hemispherical perimeter remains the energy minimum of the well, where the directional force preferentially positions the nanocrystal. We have revised the text to make the necessary clarifications.

In detail, the following changes are made to the manuscript:

1. The text on page 4 (lines 174-176) and page 5 (lines 187-189) is modified to emphasize that triangular wells are being considered.
“In the triangular geometry, it is clear that the asymmetry favors the nanocrystal positioning in the smallest angle.”
“This directionality comes from the asymmetry engineered into the meniscus with the largest effective force experienced towards the corner with the largest volume of liquid in the case of the triangular well – that is, the arc meniscus with the largest area.”
2. The text on page 7 (lines 280-282) is modified to provide further clarifications.
“In this design, as the arc length at the hemispherical perimeter is greater than that of the arc meniscus, the asymmetric meniscus design induces an effective force directed towards the rounded corner where the nanocrystal is preferentially placed.”

2. Is the nanoparticle nanocrystalline or polycrystalline? I suggest the authors provide TEM images.

As per the reviewer recommendation, we performed TEM imaging and electron diffraction measurements of the fabricated nanocrystals which confirm that the nanocrystals are single crystalline. An example diffraction pattern acquired is included below. The TEM samples were prepared using focused-ion-beam milling. We have revised the manuscript to include the results of this analysis in the main text of the manuscript and an example diffraction pattern in the Supplementary Information section.

In detail, the following changes are made to the manuscript:

1. The following text is added to the manuscript page 3 (lines 104-105):
“Using transmission electron microscopy and electron diffraction measurements as shown in Supplementary Fig. 2, we confirm that the resulting nanocrystals are single crystalline.”
2. The following figure is included in the Supplementary Information section.

Supplementary Fig. 2 | Nanocrystals are single crystalline. **a**, TEM image of nanocrystals in a triangular HSQ well cross-sectioned by focused-ion beam milling. **b**, Electron diffraction of the nanocrystal.

3. The Methods section on page 10 (lines 448-456) is updated to include the details of the TEM sample preparation and imaging.

“*Cross-sectional imaging* – Focused ion beam (FIB) cross-sectioning of the nanoLED was performed using the FEI Helios 660 (Thermo Fisher Scientific). First, protective layers of carbon (100 nm) and platinum (500 nm) were deposited onto the target device using electron beam-mediated deposition with a gas injection system. Additionally, a secondary layer of platinum (500 nm) was deposited using ion-beam mediated deposition. Cross-sectioning was then performed using gallium ions. A similar approach was used to prepare a thin lamella for TEM imaging and diffraction measurements. The lamella, which consisted of nanocrystals formed in HSQ wells on a SiO₂/Si substrate,

was imaged with JEOL ARM 200F scanning transmission electron microscope at 200 kV accelerating voltage.

3. It is suggested that the PL peak shows blueshift with nanocrystal size decrease. However, there is no data about the size of the nanocrystal. There is also no size distribution of the particles. Several words were used to mean “size”, such as size (even particle size or nanocrystal size), high, area and volume in the manuscript. It is very confusing. It is better to clearly define about the size of the nanocrystals and the particles (including that in the lateral dimensions and vertical dimension). Is the thickness of particle determined by the well thickness?

A detailed analysis of the nanocrystal size as a function of well size is included in Fig. 3. As shown in the AFM images in Fig. 3a and indicated on page 5 (lines 218-219), the nanocrystal’s lateral dimension is different from its height, and this height is set by the well thickness. As the nanocrystal lateral dimension differs from its thickness, in Fig. 3b we have presented the trend in nanocrystal size in terms of its volume. To help clarify this plot, we have now added a table to the Supplementary Information section (also included below) where the well area/thickness and nanocrystal lateral dimension/height corresponding to each point plotted is summarized. The corresponding explanation is added to page 5 (lines 221-223). This table also helps provide the necessary information to identify the size trend associated with the PL intensity plots in Fig. 3d. It should be noted that as shown in Fig. 3c, increase in the well size can also increase instances of multiple nucleation. As a result, in Fig. 3d, to facilitate an understanding of the effect of nanocrystal size on optical response, we have emphasized wells primarily leading to single nanocrystals which are highlighted in a darker color. We have changed the caption to highlight this point which previously was only mentioned on page 6 (lines 244-245). Lastly, we note that size distribution observed for each nanocrystal growth condition is reflected in the standard-deviation plotted on Fig. 3b where the results presented are average results over 30 wells. We have also made necessary changes throughout the manuscript and Supplementary Information section including in the labels and caption for Fig. 3, Supplementary Fig. 6 and Supplementary Fig. 7 so that our terminology used to refer to the nanocrystal size is consistent and clear. We have previously been using nanocrystals and nanoparticles interchangeably. We have revised the manuscript and Supplementary Information to consistently use nanocrystals in relevant places.

Supplementary Table 2 | Nanocrystal dimensions as a function of well size. The nanocrystal lateral dimensions and height associated with each growth condition presented in Fig. 3b. The dimensions are presented for wells yielding single nucleations. N/A signifies wells that did not yield a measurable fraction of single nucleations. NC length shows the mean lateral dimensions based on the lengths of the major and minor axes of the nanocrystal. Each number represents mean \pm standard deviation.

Well height (nm)	Well area (nm ²)	NC major axis length (nm)	NC minor axis length (nm)	NC length (nm)	NC height (nm)	Mean number of NCs/well
22	150 ²	42 \pm 5	34 \pm 4	38 \pm 3	21 \pm 2	1.2
	200 ²	50 \pm 5	38 \pm 3	44 \pm 2	24 \pm 2	1.5
	250 ²	N/A	N/A	N/A	N/A	2.0
28	150 ²	44 \pm 8	37 \pm 5	40 \pm 6	23 \pm 2	1.3

	200 ²	64 ± 7	42 ± 6	54 ± 6	29 ± 4	1.7
	250 ²	72 ± 9	56 ± 5	64 ± 6	36 ± 4	1.8
36	150 ²	40 ± 4	31 ± 3	35 ± 3	37 ± 2	1.5
	200 ²	N/A	N/A	N/A	N/A	2.1
	250 ²	N/A	N/A	N/A	N/A	2.3

4. Is the demonstrated technique general to other perovskite nanocrystals?

Our demonstrated technique is versatile and can be extended to other perovskite nanocrystals. To demonstrate this point, we prepared samples of organic-inorganic CH₃NH₃PbBr₃ nanocrystals. The results are included in the Supplementary Information section as Supplementary Fig. 3 (also shown below).

In detail, the following changes are made to the manuscript:

1. The following text is added to page 3 (lines 112-113).
“The process is also versatile and can be extended to other types of perovskites with an example of CH₃NH₃PbBr₃ demonstrated in Supplementary Fig. 3.”
2. The following figure, demonstrating CH₃NH₃PbBr₃ nanocrystals is added to Supplementary Fig. 3.

Supplementary Fig. 3 | On-site growth of CH₃NH₃PbBr₃ perovskite nanocrystals. SEM images of **a**, square and **b**, triangular HSQ wells containing CH₃NH₃PbBr₃ nanocrystals grown given 30° and 45° contact angles, respectively, and precursor concentration of 0.45 M. **c**, Representative photoluminescence (PL) spectrum of CH₃NH₃PbBr₃ nanocrystals formed.

5. It looks that some particles are not positioned at the desirable place. Can the authors approach the limit with high placement accuracy?

In Figs. 2a and 2d, we explore how positional accuracy scales with template geometry. As explained in Supplementary Note 2, we define “positional accuracy” as the expected distance of nanocrystals from a chosen reference point, i.e. the center of the cluster containing the greatest

fraction of nanocrystals. A smaller value of positional accuracy therefore suggests that across the population of wells, more nanocrystals were observed at the desired site. Using the particular well geometries studied in Fig. 2, we show that with increase in the well asymmetry (as template smallest angle decreases), we reach sub-50 nm positional accuracy. This number is adversely influenced by the fact that some wells contain multiple nanocrystals. In the case, where we only consider wells with single nanocrystals, a positional accuracy better than 10 nm can be realized.

We expect that increasing the degree of meniscus asymmetry, such as by increasing well asymmetry, may further improve positioning. At the extreme, though, the resolution limits of lithographic patterning, as well as the achievable degree of uniformity in surface functionalization across the population of wells, will limit positioning accuracy.

Reviewer #2

The manuscript reports a novel bottom-up template-assisted synthesis perovskite nanocrystal arrays. In brief, the authors employed topographical templates with controlled surface wettability to guide the growth and positioning of perovskite crystallites. The authors made substantial efforts to understand the underlying principles of growth and attempted to optimize the nucleation sites of the perovskite by modifying the size and shape of the templates. The authors have further demonstrated that the perovskite crystallites can be used to make high yield LED arrays. The level of control is very impressive. The study can potentially make a very nice contribution to the community and to nature communications. Below are some suggestions for further improvements:

We thank the reviewer for their comments and suggestions. We have provided our detailed response to each point below.

1. According to the current results, although the nucleation sites could be controlled within 50 nm, their morphology or orientation does not appear to be uniform, which may have contributed to the non-uniform PL and EL in the LED arrays. The authors are suggested to conduct a more detailed analysis of the size, morphology variation and their impact on the PL or EL emissions.

We agree with the reviewer that variations in size, orientation, and morphology could contribute to the nonuniformities in the PL and EL emissions. We consider the effects of particle size on PL emission in Supplementary Fig 8 in detail, which respectively show how particle size and PL emission vary across our populations as a function of well area. It should be noted that in these studies we leverage the well area as a direct approach to tuning nanocrystal size which in the studies reported in Fig 3, we establish can be achieved in a controlled manner.

However, nanocrystal morphology and orientation cannot be predetermined from the well size. As a result, to evaluate their impact on PL and EL, correlated studies of the same nanocrystal using spectroscopic and microscopic techniques would be required. Given perovskites' sensitivity to environmental conditions, the samples used for spectroscopic characterization in our current studies are prepared inside a nitrogen glovebox and immediately encapsulated with a polymer layer prior to performing the spectroscopy in ambient, as described in the Methods

section (lines 375-378). The encapsulant, which prevents potential degradations or alternation in properties due to uncontrollable changes in the environmental conditions or under laser excitation, is necessary to most effectively measure and report the optical response of the nanocrystals. This polymer-coated sample, though, is not amenable to high-resolution SEM imaging needed to decipher the crystal's morphology or orientation. More advanced spectroscopy setups with environmental control would accommodate such measurements without a need for an encapsulant which could be followed by electron microscopy. This task, which is beyond the scope of our current manuscript, will be part of our future studies with access to the appropriate measurement platform. Indeed, enabling such correlated high-throughput measurements, given the appropriate metrology platforms, is a unique feature enabled by our reported on-site growth technique. This would not be limited to studies of size, orientation, morphology and can be performed over a diverse chemical and physical design space including perovskite's composition, ligands for surface passivation, exposure to diverse environmental factors, etc. These are currently some of the topics of our ongoing work and we hope that they provide other opportunities for the community to collectively study and identify the structure-chemistry-function relationships in halide perovskites and their effects on the optoelectronic applications.

We have revised the manuscript to add the following paragraph on page 6, lines 252-258:

“In allowing scalable, controlled formation of perovskite nanocrystals, our platform provides a high-throughput approach to examine how the nanocrystals' physical and chemical features affect their optoelectronic properties. For example, the effects of nanocrystal size, orientation, and morphology on PL emission may be elucidated, which are speculated to contribute to the current nonuniformities observed in the emission characteristics. Furthermore, nanocrystal composition, ligands, surface passivation, and growth conditions may be varied to optimize the performance of the nanocrystals, whose photoluminescence quantum yield is estimated in Supplementary Note 5.”

2. The EL image, although with a very impressive yield, showed a relatively high degree of nonuniformity. Additionally, there is no discussion of the EL efficiency either. For LED display applications, the uniformity of the pixels is a critical factor that must be considered. It is important for the authors to explore the underlying reasons for such non-uniformity and the possible strategies to address the problem.

We agree with the reviewer that minimizing inter-pixel variation in emission is important for nanoLEDs to be applied in displays, as well as in applications requiring on-chip light sources, like optical communication, computation, or quantum light sources.

The electroluminescence external quantum efficiency (EQE) relates the photon emission rate to the electron injection rate. Here, we calculate the EQE by relating the optical power [Watts] of the LEDs calculated and reported in Supplementary Fig. 13 to the energy [Joules] of each emitted photon calculated from the peak wavelength of the EL spectrum to determine the number of photons emitted per unit time; while the number of electrons injected per unit time is determined from the device current (Supplementary Fig. 9a). Assuming all the current we measure passes through our nanoLEDs (600 LEDs on the given electrically-probed pad), the

resulting average EQE of a nanoLED is $\sim 5 \times 10^{-10}$. The details corresponding to EQE estimation is included in Supplementary Note 6.

Given all LEDs per electrically-probed pad share common ITO and Al electrode, a poorly-formed nanocrystal or a resistive-short in one well can affect the entire pad. In particular, such shorts can sink significant current, and therefore limit the amount of current that passes through functional devices to produce light. This in turn can greatly affect the ratio of photons to electrons (EQE).

Furthermore, beyond defects within a well, one should also consider the space between wells all of which share common ITO/Al layers. In our demonstration, each of our LEDs has an active area of $1.23 \mu\text{m}^2$. However, to enable facile device probing, large electrodes are utilized for contact, so that the total area between each top and bottom electrode is 5.44 mm^2 . With 600 LEDs per electrode-pair, the fraction of LED area to electrode area is 1.36×10^{-4} . The large, unutilized areas may include defects that can also increase current draw via resistive shorts. To limit such losses, as a next step in this work, it would be valuable to develop individually addressable nanoLEDs with reduced electrode sizes.

Finally, being bottom emitting devices (light comes out via the glass substrate), there are some outcoupling losses which we are not taking into account in our estimation.

In detail, the following changes are made to the manuscript:

1. The following text is added to page 7 (lines 300-302).
 “The details of this estimate are provided in Supplementary Note 6, along with an analysis of the external quantum efficiency.”
2. The following text is added to Supplementary Note 6, page 13.

The external quantum efficiency (EQE) of a single nanoLED can be calculated as the ratio of the photon emission rate to the electron injection rate. Assuming all the current we measured passes through our nanoLEDs, the averaged EQE can then be expressed as:

$$\eta_{\text{EQE}} = \frac{P_{\text{ph}}}{h\nu} / \frac{I}{e n} \quad (30)$$

where P_{ph} is the total photon emission rate from the given electrically-probed pad, I is the total current injected from Supplementary

Fig. 9a, and $\bar{\cdot}$ stands for average. We therefore have $\eta_{\text{EQE}} \approx 5 \times 10^{-10}$.

3. Lastly, although it is claimed as EL, there is no evidence to support whether the emission is from direct EL from the perovskite crystallites or down-conversion of EL from the TPBi layer. Although this may not affect the apparent device function, it could affect the overall efficiency. The authors are encourage to look into or at least discuss about relevant uncertainties.

Owing to the fact that the EL emission spectrum observed only corresponds to the perovskite nanocrystals (peaking at ~520 nm, without a signature of the TPBi emission with a peak expected at ~440 nm), as shown in Supplementary Fig. 9b, we believe the emission mechanism in these devices is electroluminescence from the perovskite nanocrystals themselves rather than down-conversion of EL from the TPBi layer. Extending on this, owing to the relatively large size of the well in comparison to the lateral dimensions of the particle, it would be difficult to expect complete down-conversion or energy transfer of excitons into the perovskite without any spectral signature of the TPBi in the EL spectrum further reinforcing our inference that what we observe is EL emission from the nanocrystals themselves.

The inclusion of a thin-PMMA blocking layer between the ITO electrode and the TPBi transport layer, should help limit injection of both charges into the TPBi layer. Similar use of PMMA to limit background emission from charge transport layers in quantum-dot based single-photon sources has been reported in the literature (Citation #37 in main text, DOI: 10.1038/s41467-017-01379-6) and also used in Citation #41, DOI: 10.1021/acsnano.2c00488.

Though it does not seem to be the primary mode of operation in the devices, we do agree with the reviewer that if such a down-conversion mechanism is present in the device this could further impact the overall efficiency of the device by expending energy in forming higher energy (blue) excitons in the TPBi to ultimately emit light of lower energy (green) from the perovskite nanocrystal.

Reviewer # 3

The emerging perovskite materials show great promises for the next generation micro-LED. However, most perovskite LEDs are based on polycrystalline films prepared by spin coating, which need sophisticated lithography steps to get micro-LED. The pattern of nanocrystals is an alternative strategy. But traditional patterning is limited to micron-sized patterning, and the sub-lithographic precision cannot be guaranteed as the patterning area is confined by the precursor solution. Here, in this manuscript the authors reported the integration of perovskite nanocrystal arrays into nanoscale LEDs. Both the size and position of crystals can be well controlled down to <50 nm. I believe the further optimization of this kind of sample preparation methods, their properties, and the integration into nanoscale LEDs will bring in a revolution for LEDs. The developed method in this manuscript will also be beneficial to other on-chip nanodevices. Here are some questions for the improvement of this manuscript:

We thank the reviewers for their comments. The detailed answers to their questions are provided below.

1. What's the PLQY of the crystals? As well known, the efficiency will decrease as the size of LED decreases. What's the EQE of this device? Could the authors show these data?

As per the recommendation of the reviewer, we measured and estimated the PL quantum yield of the nanocrystals and the details are included in Supplementary Note 5. As suggested, a reduction in the PLQY is observed with decrease in the crystal size. The measured values are estimated to

be ~ 10% - 20%. However, we expect that through means of surface passivation and controlled growth environments, the optical response and efficiency can be improved.

The electroluminescence external quantum efficiency (EQE) relates the photon emission rate to the electron injection rate. Here we calculate the EQE by relating the optical power [Watts] of the LEDs calculated and reported in Supplementary Fig. 13 to the energy [Joules] of each emitted photon calculated from the peak wavelength of the EL spectrum, to determine the number of photons emitted per unit time; while the number of electrons injected per unit time is determined from the device current (Supplementary Fig. 9a). Assuming all the current we measure passes through our nanoLEDs (600 LEDs on the given electrically-probed pad), the resulting average EQE of a nanoLED is $\sim 5 \times 10^{-10}$. The details corresponding to EQE estimation is included in Supplementary Note 6.

Given all 600 LEDs per electrically-probed pad share common ITO and Al electrode, a poorly-formed nanocrystal or a resistive-short in one well can affect the whole pad. In particular, such shorts can sink a lot of the current, and therefore limit the amount of current actually passing through functional devices to produce light. This in turn can significantly affect the ratio of photons to electrons (EQE).

Furthermore, beyond defects within a well, one should also consider the space between wells all of which share common ITO/Al layers. In our demonstration, each of our LEDs has an active area of $1.23 \mu\text{m}^2$. However, to enable facile device probing, large electrodes are utilized for contact, so that the total area between each top and bottom electrode is 5.44 mm^2 . With 600 LEDs per electrode-pair, the fraction of LED area to electrode area is 1.36×10^{-4} . The large, unutilized areas may include defects that can also increase current draw via resistive shorts. To limit such losses, as a next step in this work, it would be valuable to develop individually addressable nanoLEDs with reduced electrode sizes.

Finally, being bottom emitting devices (light comes out via the glass substrate), there are some outcoupling losses which we are not taking into account in our estimation.

In detail, the following changes are made to the manuscript:

1. Supplementary Note 5 on PLQY measurements and estimation is added.
2. The following text is added to page 7 (lines 300-302).
“The details of this estimate are provided in Supplementary Note 6, along with an analysis of the external quantum efficiency.”
3. The following text is added to Supplementary Note 6, page 13.

The external quantum efficiency (EQE) of a single nanoLED can be calculated as the ratio of the photon emission rate to the electron injection rate. Assuming all the current we measured passes through our nanoLEDs, the averaged EQE can then be expressed as:

$$\eta_{\text{EQE}} = \frac{P_{\text{opt}}}{h\nu} / e n \quad (30)$$

where e is the elementary charge, $n = 600$ is the number of nanoLEDs on the given electrically-probed pad, $\approx 3 \times 10^7 \text{A}$ is the total injection current from Supplementary Fig. 9a, and $\bar{\cdot}$ stands for average. We therefore have $\eta_{\text{eqe}} \approx 5 \times 10^{-10}$.

2. As shown in Fig. 1c and Fig. 4d, for both PL and EL, the emission intensity is not homogeneously across the whole patterns. For some area, the emission intensity is rather low. What's the reason for that, can it be improved?

The nonuniformities in the PL and EL emission might in part result from variations in size, morphology and orientation of the nanocrystals. We have comprehensively studied the size variations. The effect is discussed in the manuscript and the Supplementary Information section. Such study is feasible since we can systematically tune the nanocrystal size by design through changing the template size. Such tuning of the morphology and orientation however is not currently feasible. As a result, studying their effects requires correlated electron microscopy and optical spectroscopy measurements on the same nanocrystal. To maintain the stability of the nanocrystals, the samples in our spectroscopic studies are encapsulated in a polymer layer upon formation in a glovebox which would in turn not allow electron microscopy measurements on the same samples. Future development of measurement setups that allows measurements in an inert/controlled environment would alleviate this limitation to help elucidate the effect of morphology and orientation on the emission characteristics. To this end, given that our approach allows high-throughput nanocrystal processing, it provides a platform to explore a diverse chemical and structural design space to understand the structure-chemistry-function relationships in perovskite nanocrystals towards development of the material properties for desired applications. These include the effect of size, morphology, orientation, surface passivation, type of ligands, amongst other factors.

We have revised the manuscript to add the following paragraph on page 6, lines 252-258 and summarize some causes of variations in the optical response.

“In allowing scalable, controlled formation of perovskite nanocrystals, our platform provides a high-throughput approach to examine how the nanocrystals’ physical and chemical features affect their optoelectronic properties. For example, the effects of nanocrystal size, orientation, and morphology on PL emission may be elucidated, which are speculated to contribute to the current nonuniformities observed in the emission characteristics. Furthermore, nanocrystal composition, ligands, surface passivation, and growth conditions may be varied to optimize the performance of the nanocrystals, whose photoluminescence quantum yield is estimated in Supplementary Note 5.”

3. The architecture of the LED should be optimized. Currently, the crystal is sandwiched between 5nm PMMA and TPBi, there is no balanced injection of electrons and holes. The recombination of electrons and holes in the crystals may also not be efficient.

Here, we have utilized a hole-transport layer free LED architecture, also commonly reported by others ([H. Zhang, et al. *ACS Nano* 16, 6394-6403, 2022]; [Yuan, F. et al. *Frontiers in Chemistry* 10, 1-9, 2022]) to demonstrate the potential of our platform for scalable fabrication of deterministic nanoLEDs. Similar to these reports, our thin-PMMA blocking layer may allow hole

accumulation within our nanocrystal,^{2,3} which due to its bipolar injection characteristics, acts as a HTL in addition to an emitting layer.²

We do agree with the reviewer that other architectures including those with a distinct HTL may improve efficiency. Having shown a new platform for making perovskite nanoLEDs in our manuscript, our technique can be used broadly with other perovskites and device geometries to realize nanoLEDs with improved efficiencies or other functionalities depending on the desired application. We have added a note about this to the manuscript on page 7 (lines 305-307):

“Additionally, by controlling the nanocrystal composition and surface passivation, or utilizing other device architectures, LEDs with tunable emission wavelength and improved efficiencies may be achieved.”

0. In Fig. 4b, the arc crystals only cover a few area between PMMA and TPBi. However, if the authors use cubic crystals, the coverage may be better, and the EL performance may also be improved.

We thank the reviewer for their comment. In this work, we have demonstrated how tuning surface forces enables precise control over nanocrystal size and positioning. We have not explored whether this technique can be leveraged to control nanocrystal morphology. While we expect that some degree of morphological control may be achievable through optimization of precursor ratios, solvent mixtures, evaporation conditions and well geometry, we believe demonstrating control over this parameter is out of the scope for our current work.

As noted by the reviewer, in the current design, the nanocrystal within the teardrop well covers a fraction of the well area. Indeed, this is a unique feature of our approach where sub-lithographic resolutions in defining the nanocrystal size are enabled despite using lithographic techniques for the fabrication of the wells. This allows overcoming the resolution limits of lithography to achieve conventionally inaccessible dimensions. Based on our studies, the added PMMA layer in our LED design limits the leakage current and TPBi background emission as noted in the manuscript and also shown by others in the literature included as references [37] and [41] in our manuscript. It is however possible that optimization of the nanocrystal coverage and well geometry can further improve the EL emission and will be subjects of future studies.

1. Electrodes of thin-film transistor (TFT) can be precisely controlled into nanoscale. TFT architecture may be more suitable for this small crystal.

We thank the reviewer for their comment. Indeed, an advantage of our approach is that it can be easily processed over pre-fabricated features including those of a TFT electrodes. In this design, the TFT features can be leveraged to serve as the template for nanocrystal positioning and growth enabling a self-aligned process for nanoLED fabrication already integrated with a TFT. This would allow addressing each nanoLED individually.

2. The authors said they got the first perovskite nanoLED. As I know, there was another report of nanoLED (Nature Photonics, 2022: 16, 284-290).

We thank the reviewer for bringing this paper to our attention. We have now included this in our list of cited references in the manuscript (new reference [39]). A notable difference between our approach and that reported in this paper is that our approach enables formation of perovskites nanocrystals (thus nanoLEDs) with deterministic spatial order and nanometer placement accuracy on the level of individual nanocrystals while the referenced paper relying on stochastic nanopores formed inside an aluminum membrane is not compatible with deterministic nanoLED fabrication for on-chip device integration. The figure below helps compare the two approaches to highlight and clarify this difference.

(a) Porous aluminum membranes with stochastically placed pores are used to form arrays of uniform crystalline perovskite quantum wires that are formed into LEDs larger than millimeters in size (Nature Photonics, 2022: 16, 284-290). (b) Deterministically fabricated perovskite nanocrystals with single nanocrystal control and nanometer placement accuracy formed into deterministic nanoLEDs. Each nanocrystal can be formed into a nanopixel (this work) and hence providing opportunities for on-chip integrated nanodevices.

References

1. Zhang, H. *et al.* High-Luminance Microsized $\text{CH}_3\text{NH}_3\text{PbBr}_3$ Single-Crystal-Based Light-Emitting Diodes via a Facile Liquid-Insulator Bridging Route. *ACS Nano* **16**, 6394–6403 (2022).
2. Yuan, F. *et al.* Hole Transport Layer Free Perovskite Light-Emitting Diodes With High-Brightness and Air-Stability Based on Solution-Processed CsPbBr_3 - Cs_4PbBr_6 Composites Films. *Front. Chem.* **10**, 1–9 (2022).
3. Shi, Y. *et al.* A Strategy for Architecture Design of Crystalline Perovskite Light-Emitting Diodes with High Performance. *Adv. Mater.* **30**, 1–10 (2018).

REVIEWERS' COMMENTS

Reviewer #1 (Remarks to the Author):

As the authors addressed all my concerns, I suggest it can be accepted for publication.

Reviewer #2 (Remarks to the Author):

The authors have sufficient addressed my questions and the manuscript is now acceptable for publication.

Reviewer #3 (Remarks to the Author):

The authors replied all my questions correctly. I suggest the acceptance of this manuscript.